# The Role of Oxidative Stress as a Mechanism in the Pathogenesis of Acute Heart Failure in Acute Kidney Injury

**DOI:** 10.3390/diagnostics14182094

**Published:** 2024-09-23

**Authors:** Danijela Tasić, Zorica Dimitrijević

**Affiliations:** Clinic of Nephrology Prof Dr Spira Strahinjić, University Clinical Center Niš, Faculty of Medicine, University of Niš, 18000 Niš, Serbia; zorica_mdimitrijevic@yahoo.com

**Keywords:** oxidative stress, acute heart failure, cardiorenal syndrome

## Abstract

Despite a large amount of research on synchronous and mutually induced kidney and heart damage, the basis of the disease is still not fully clarified. Healthy mitochondria are essential for normal kidney and heart function. Mitochondrial dysfunction occurs when the clearance or process of generation and fragmentation of mitochondria is disturbed. The kidney is the second organ after the heart in terms of the number of mitochondria. Kidney tubules are rich in mitochondria due to the high energy requirements for absorption of large amounts of ultrafiltrate and dissolved substances. The place of action of oxidative stress is the influence on the balance in the production and breakdown of the mitochondrial reactive oxygen species. A more precise determination of the place and role of key factors that play a role in the onset of the disease is necessary for understanding the nature of the onset of the disease and the creation of therapy in the future. This underscores the urgent need for further research. The narrative review integrates results found in previously performed studies that have evaluated oxidative stress participation in cardiorenal syndrome type 3.

## 1. Introduction

Acute kidney injury is a complex clinical syndrome. A general definition of acute kidney injury (AKI) says that it is an abrupt decrease in functional reserves of glomeruli resulting in increased nitrogenous waste products in persons with an average anatomic quantum of healthy renal tissue. According to the last nomenclature for kidney function and disease, a definition and classification of the Clinical Practice Guideline for Acute Kidney Injury (AKI) and Acute Kidney Disease (AKD) in the Kidney Disease Improving Global Outcomes (KDIGO) guideline from 2012 is still accepted, and according to it, acute kidney injury is a subgroup of acute kidney diseases. It is an abrupt decrease in the intensity of glomerular filtration, manifested and determined by the increase in serum creatinine and different urine volume or oliguria as available tools in everyday clinical practice. The role of the approved biomarkers in clinical practice in different resource settings is not yet evident according to nephrological clinical guidelines, nor is it available in everyday nephrology practice. The time during which renal function decreases is different, and it is measured in hours and days [1]. The occurrence of acute kidney injury is associated with a prolonged stay during hospital treatment, high mortality, and increased cost of treating patients [2,3,4,5]. In addition, the onset, severity, number of episodes, duration, and outcome of acute kidney function loss are affected by the underlying disease, comorbidities, altered pharmacodynamic profile of drugs, and risk factors [6,7]. Due to the frequent occurrence of multiorgan dysfunction in acute kidney injury, the involvement of the scientific and professional public is primarily dedicated to understanding the connection between the kidneys and other organs [8,9,10]. Heart failure is a clinical syndrome consisting of cardiac symptoms that may be accompanied by signs” [11]. It is a structural and/or functional abnormality of the heart that prevents normal inflow or undisturbed emptying of the ventricles, whereby circulatory failure is not a mandatory phenomenon. Acute decompensated heart failure caused by the sudden development of renal injury is a life-threatening condition whose main clinical feature, in addition to cardiogenic shock and acute pulmonary edema, may also be severe arrhythmias [11]. Acute cardiorenal interaction is based on hemodynamic and non-hemodynamic mechanisms and metabolic abnormalities that, together with other mechanisms, accelerate the functional damage of both organs [12,13,14,15].

AKI is not only an indicator of the severity of the disease but also activates other mechanisms that play a leading role and cause long-term consequences on the function of other organs and systems [16,17,18,19,20]. It became clear that the universal clinical use of biomarkers in different forms of AKI is not possible because each etiopathogenetic mechanism leads to different molecular, cellular, and functional changes [21,22,23,24,25,26].

The fact is that all clinical manifestations of AKI cannot be explained by one pathophysiological mechanism [27,28,29]. One of the basic pathophysiological mechanisms in AKI that affects the damage to other organs is oxidative stress.

Oxidative stress has been identified as a pathogenetic mechanism in various acute and chronic diseases, including primary and secondary kidney and heart diseases. The result is an imbalance between the creation of reactive oxygen species (ROS) and the endogenous antioxidant capacity [30,31,32]. It plays a crucial role in primary heart and kidney disease mechanisms. The capacity of oxidative stress to introduce the affected cells in the body into processes from apoptosis to necrosis is realized with the help of ROS, which, although created only from one source, are capable of initiating a series of new reactions with short-term and long-term consequences [33,34,35,36,37,38,39].

Current research aims to diagnose numerous forms of AKI more precisely (such as subacute, subclinical, transitory, prolonged, or severe), apply treatment according to the initial lesion, and prevent potential damage to other organs.

Bearing in mind that AKI is not just a naive observer but an actor in acute disease, this narrative review deals with the current topic related to cardiovascular consequences caused by local and systemic oxidative stress during AKI.

### 1.1. Oxidative Stress and Mechanisms of Its Occurrence

Oxidative stress occurs due to a disturbed balance between oxidant activity, i.e., the creation of free radicals, and the function of the antioxidant system in the body. Damage to the structure of the cell and disruption of its function occurs during a large accumulation of oxidants and antioxidants. This imbalance plays a major role in the oxidative damage of the cell, on the molecules of proteins, lipids, carbohydrates, enzymes, and nucleic acids, in the role of oxidases, reactive oxygen species (ROS) formed in these reactions, which can be used as biomarkers of oxidative stress. ROS in small concentrations is essential for maintaining homeostasis at a given moment. A slight increase in ROS and reactive nitrogen species (RNS) levels, responsible for normal redox signaling, promotes cell proliferation and differentiation [40,41]. Oxidative stress can directly affect deoxyribonucleic acid (DNA) and initiate mutagenic lesions, and indirectly, by initiating autocatalytic lipid peroxidation, it can influence the formation of DNA adducts. During lipid peroxidation, which is the most significant negative consequence of synthesizing ROS, genotoxic decomposition products are created, of which the indicator of the level of lipid peroxidation is Thiobarbituric Acid Reactive Substance (TBARS), thiobarbituric acid and its end product [42,43]. It has been shown that oxidized proteins, especially products of oxidative modification of protein Advanced Oxidation Protein Product (AOPP), are created during the activation of oxidative activity at the level of neutrophils and monocytes. Stimulation of these cells causes the so-called monocyte respiratory burst and the secretion of various inflammatory mediators. Their measurement is also a marker of monocyte activation (Figure 1) [44,45].

The final and most mutagenic lipid peroxidation product is malonyl dialdehyde (MDA). It is a marker of oxidative stress that indicates the degree of cell membrane damage induced by free radicals. Intense lipid peroxidation of cell membranes leads to loss of turgor of cell membranes, reduction of membrane potential, and increase of their permeability to hydrogen ions, leading to cell rupture [46,47]. Thiols are the main components of extracellular non-enzymatic antioxidant mechanisms. Thiol (R-SH sulfhydryl group) is an organic sulfur compound that can remove free radicals. A reduced concentration of free thiol groups enables oxidative stress to be maintained. Thiols are a biomarker for systemic reduction status and degree of systemic oxidative stress. Other antioxidant compounds such as glutathione, homocysteine, and cysteine contain low molecular weight (LMW) thiols and are less important for antioxidant capacity. Free thiols have the largest share because a high level of systemic free thiols indicates a more favorable redox status. Plasma proteins contain the largest amount of redox-active thiol groups. Thiols are a biomarker for identifying patients at risk of progression and development of acute kidney injury (AKI). Thiols, as a pathophysiological indicator, are key transduction elements in redox signaling, which is why they are recognized as indicators of oxidative stress. Research has shown a correlation between low levels of thiols and AKI, which confirmed once again that oxidative stress plays an important role in the pathophysiology of AKI (Figure 2) [48,49,50,51].

### 1.2. Oxidative Stress in Patients with AKI

In the kidneys, the blood flow and glomerular filtration rate are maintained by autoregulation, myogenic response, and tubuloglomerular feedback, all under the influence of nitrogen oxide (NO) and superoxide anion (O_2_^−^) [52]. These vasodilator molecules protect renal endothelial and mesangial cells from apoptosis and fibrosis by stimulating antioxidant genes, while Nitric oxide synthase (NOS) inhibition reduces renal blood flow and increases renal vascular resistance even at concentrations that do not affect blood pressure [53]. Interestingly, ROS produced by mitochondria can have a renoprotective effect by moderately increasing the concentration of hypoxia-inducible factors (HIF) in endothelial cells and stimulating the synthesis of the main regulator of the oxidative stress response (NpF_2_) nuclear factor erythroid 2-related factor 2 [54,55]. This underscores the body’s intricate defense mechanisms against oxidative stress.

The main factor for maintaining physiological tone in the blood vessels of the kidneys, endogenous O_2_, is induced by Nicotinamide adenine dinucleotide phosphate (NADPH) oxidase (NOX) in the blood vessels [56,57]. NADPH oxidase and xanthine oxidase are enzymes that are involved in the occurrence of oxidative stress. In polymorphonuclear leukocytes, under the action of NADPH oxidase, free radicals are created during the process of converting oxygen molecules into superoxide anions [58]. Further, the superoxide anion is converted into hydrogen peroxide H_2_O_2_. Both products of NADPH oxidase activity, superoxide anion and hydrogen peroxide, are precursors for producing more potent oxidants by creating conditions for other reactions [59]. Nitrosative stress occurs when the superoxide anion reacts with nitrogen oxide; on this occasion, the toxic product peroxynitrite is created [60].

The reaction is known as the Fenton reaction of classic oxidative stress when hydrogen peroxide and intracellular iron react. The Haber Weiss reaction occurs when superoxide anion and hydrogen peroxide interact, producing the hydroxyl radical [61].

Chlorinated stress occurs when hydrogen peroxide in the presence of chlorine, and under the action of myeloperoxidase (MPO) from polymorphonuclear leukocytes, turns into hypochlorous acid, which reacts with endogenous amines, creating chloramines [62]. Degradation of arachidonic acid in the process of lipid peroxidation by enzymatic and non-enzymatic reactions produces malonyl dialdehyde (MDA). Once created, it undergoes oxidation under the influence of enzymes from the mitochondria or reacts with proteins on the cell membrane or DNA, causing mutagenic damage. The concentration of MDA in the urine is directly proportional to the degree of kidney tubule damage, which is why it can be a good indicator of tubule damage in acute renal failure [63,64]. Processes of stimulated lipid peroxidation increase the risk of ischemia-reperfusion damage. An increase in MDA concentration is not only a sign of activity but also a consequence of the accumulation of oxidative stress molecules that damaged kidneys cannot excrete [64,65].

### 1.3. The Role of Mitochondria in Acute Cardiorenal Syndrome

Undoubtedly, mitochondria’s role in acute renocardial syndrome is pivotal. Comprising a third of the cell’s volume, they play a decisive role in cellular organelle interactions, highlighting the interconnectedness of cellular processes. Their energy production nearly meets the body’s needs, underscoring their significance in this syndrome.

The antioxidant capacity of mitochondria reduces H_2_O_2_ to water and thus physiologically ensures a low level of ROS. In disturbed conditions, ROS are produced in larger quantities and exceed the antioxidative capacity of the organelle. As a consequence of accumulation, mitochondrial ROS (mtROS) rapidly reacts with NO, releasing H_2_O_2_ into the cytoplasm and forming an environment for nitrification and worsening oxidative damage outside the mitochondria [66,67].

The crux of maintaining cell vitality and mitochondrial function lies in the delicate and intricate balance between the production and elimination of mtROS. This balance is not just a key factor, but it is the key to understanding mitochondrial function. Once disturbed homeostasis in mitochondria caused by oxidative stress further increases the production of ROS and leads to the so-called “ROS induced ROS to release” vicious circle, disruption of mitochondrial function and morphology, impaired mitochondrial wall permeability, and cell death. Reactive oxygen species stress increases the tendency for mutations that reduce ATP synthesis, leading to the inactivation of the calcium channel pump and a consequent increase in intracellular calcium, while activating phospholipase promotes the breakdown of phospholipids on the membrane [68,69].

Mitochondrial dysfunction is the basis of kidney and heart diseases. The most important event in kidney damage during AKI is tubule cell apoptosis, which leads to mitochondrial fragmentation, resulting in reduced energy metabolism and increased ROS generation. An increased amount of mtROS affects the function of kidney and heart cells in acute renocardial syndrome. Mitochondrial morphological changes, fission, and fusion are mediated by guanosine triphosphatases (GTPases). The fusion process of the outer membrane is regulated by mitofusin 1 (MFN1) and mitofusin 2 (MFN2), while the fusion of the inner membrane is controlled by optic atrophy 1 (OPA1). There is a direct connection between the morphology of mitochondria and the energy state of the cell. Mitochondrial fragmentation is dependent on MFN2 expression. Greater expression of mitofusin 2 stimulates mitochondrial oxidative phosphorylation (OXPHOS) reactions in mitochondrial metabolism. A decrease in the activity of the effector OPA1 promotes mitochondrial fragmentation and reduces oxygen consumption. Fission is regulated by protein 1, which is related to dynamin-related protein 1 (DRP1); otherwise, it is a cytosolic protein that activates the mitochondrial membrane. Activation of DRP1 in damaged mitochondria of kidneys and/or the heart during acute kidney injury plays an important role in their dysfunction Table 1 [70,71,72,73,74,75,76,77].

Impaired mitochondrial fission 1 protein (FIS1) is directly caused by the impaired function of mitochondrial dynamic proteins. The consequence of this disorder is the accumulation of mitochondrial double-stranded deoxyribonucleic acid (mtDNA) mutations and, consequently, damaged proteins. In addition, the parts of mitochondria produced by fission produce larger amounts of reactive oxygen species that damage cells with oxidative stress. The size of mitochondrial fragments is related to the performance of mitochondrial oxidative phosphorylation (OXPHOS) [78].

In damaged heart cells, gene ablation, increased protein expression, and insufficient function of inhibitory mechanisms lead to an increase in the number of mitochondria, alterations in their shape, apoptosis, and severe contractile damage of cardiomyocytes [79]. 

Increasing the concentration of intracellular nicotinamide adenine dinucleotide (NAD+) and the enzyme responsible for its production, nicotinamide mononucleotide adenosyl transferase, can help stabilize the structure of mitochondrial cristae, enhance mitochondrial respiratory function, reduce ROS release, induce apoptosis in heart cells, and improve heart function [80,81].

### 1.4. Mechanisms of Mitochondrial Dysfunction in Cardiorenal Syndrome Type 3

Mitochondria, the powerhouses of the cell, play a major role in cell function by generating adenosine triphosphate (ATP) through oxidative phosphorylation of fatty acids. They adapt to the needs of the cell and the extracellular space by forming a network of mitochondria that changes in number and appearance [82,83,84].

Fusion of mitochondria serves to keep themselves healthy and increase their capacity for oxidative phosphorylation. Deficiency of mitofusin 2 (MFN2) in the proximal epithelial cells of the kidneys caused an exceptional susceptibility to apoptosis and a significant fragmentation of mitochondria. In adults, the deficiency of mitofusin 1 (MFN1) and 2 leads to hypertrophy and dilatation of the heart muscle [85,86].

Mitochondria fission removes damaged mitochondria and changes the arrangement of the contents in the new cell created during mitosis. Early renal tubule cell damage is manifested by dyamin-related protein 1 (DRP1) activation in cisplatin-induced nephrotoxicity. In diabetics, the main contribution to the occurrence of oxidative stress is the upregulation of DRP1, while the downregulation of DRP1 improves the morphology of mitochondria and reduces the apoptosis of kidney cells. After unilateral ureteral obstruction, downregulation of DRP1 improves mitochondrial function and reduces the proliferation of fibroblasts by hypoxia-stimulated transforming growth factor beta 1 (TGFβ1), a key inducer of fibrosis in the renal proximal tubules. Mitochondrial fission is important for heart muscle function. Increased expression of DRP1 and MFN1 was found in the heart muscle of patients with heart failure, while overexpression of MFN2 in an experimental model slowed endothelial dysfunction and the initiation of atherosclerosis. Inhibition of DRP1 in cardiac muscle reduces mitochondrial fragmentation and myocardial infarct volume [87,88,89].

In the human body, the heart, blood vessels, and kidneys are the organs with the largest number of mitochondria because they need the largest amount of energy for their function.

During acute kidney and acute heart damage, the cells of these organs are insufficient for the oxidation of fatty acids, so ATP is depleted. Lipids accumulate in the cell, and an alternative path in the tubule cells in the kidneys for energy restoration and ATP synthesis is achieved with the help of enzyme glycolysis. In contrast, this modification in the heart has yet to be sufficiently researched for the time being [90,91,92].

Mitochondrial reactive oxygen species (mtROS) are removed in mitochondria using enzymatic and non-enzymatic reactions. Superoxide anion (O_2_^−^) is converted into hydrogen peroxide (H_2_O_2)_ by superoxidase dismutase (SOD). The main roles in mitochondria for reducing the concentration of hydrogen peroxide (H_2_O_2_) are played by mitochondrial catalase and glutathione peroxidase [93,94,95].

Preclinical research has shown that acute kidney damage affects the metabolism in heart muscle cells through signaling pathways that are a consequence of mitochondrial function and cause disruption of the metabolism of pyruvate, glyoxylate, dicarboxylic acid, starch, sucrose, and amino acid synthesis. The main regulator of myocardial damage in acute kidney damage is growth factor receptor binding protein 2 (Grb2). Activation of this protein or the use of Grb2 inhibitors promotes inhibition of the serine/threonine protein kinase/mammalian target of Rapamycin (Akt/mTOR) signaling pathway and thus leads to disruption of mitochondrial function in cardiomyocytes [96,97,98,99].

The idea of hormesis was presented for the first time in the 16th century in connection with the bimodal response of cells to external factors, while the term mitohormesis was introduced in the year 2006 for the response of mitochondria in subepithelial cells to a dose of stress. The knowledge that most proteins in mitochondria are coded via nuclear DNA pointed to the importance of retrograde signaling from mitochondria and the adaptive response of the nucleus for mitochondrial recovery [100,101,102,103,104]. In the cells, signal molecules inform the cytosolic system about stress in the mitochondria, so the accumulation of proteins in the cell’s cytosol activates the transcription mechanisms that are part of the mitohormetic response. The most important signaling pathway of mitochondrial stress in the cytosolic signaling system of the cell is ROS (reactive oxygen species) from mitochondria [105,106,107,108,109]. High concentrations of mitochondrial ROS are harmful, while low concentrations are a stimulus for numerous signaling pathways (Figure 3).

Maintaining mitochondrial homeostasis improves kidney function [110]. Mitohormetic pathways can be blocked by antioxidants, while DRP1 inhibitors block mitochondrial fragmentation and apoptosis to protect the kidney from ischemia and acute kidney damage caused by cisplatin [111,112]. Inhibition of mitochondrial fission and stimulation of mitochondrial fusion can be a therapeutic goal in the treatment of atherosclerosis [113,114,115].

## 2. Discussion

Numerous research studies investigate the damage to various kidney structures and the consequent functional disorders caused by oxidative stress. In addition, oxidative stress is one of the non-traditional cardiovascular risk factors responsible for disease progression and pathogenesis of the cardiorenal axis. However, despite detailed examination and numerous new findings that are sometimes controversial, suitable therapeutic approaches for the prevention and treatment of acute cardiorenal syndrome still need to be established. Perhaps the answer lies in different mechanisms of the action of oxidative stress and antioxidant enzymes. Oxidative stress manifests its effects in the kidneys according to the principle of hormesis.

Mitochondria are responsible for maintaining homeostasis and are the cell’s primary source of oxidative stress. They play an essential role in maintaining metabolic signaling in the cell and regulating the internal pathway of apoptosis. Mitochondria fulfill their complex roles because they are dynamic and heterogeneous organelles, differing in shape, size, mass, dynamics, turnover, metabolic activity, and membrane potential within the cell. Kidneys are metabolically active organs. Different parts of the nephron have a different number and arrangement of mitochondria.

Kidney tubule cells are best supplied with mitochondria because they need much energy for the processes of resorption and secretion with the help of active transport. Podocytes also need a lot of energy to take up filtered proteins and maintain the stability of the cytoskeleton proteins and the extracellular matrix. That is why mitochondria were examined not only as a risk factor in the pathogenesis of various kidney and heart diseases but also as their role in regulating oxidative stress and the possibility of treatment. Mitochondrial fission is important for mitochondrial proliferation after cell mitosis and the removal of mitochondria by mitophagy. Mitochondrial fission inhibitor-1 is important for selective, reversible inhibition of the fission protein DRP-1, which was used for the pharmacological testing of drugs with a reno-protective effect in acute cardiorenal syndrome [116].

The focus of molecular pharmacology research was mitochondrial permeability transition pore inhibitors (mPTP), which have the potential to suppress apoptosis of heart and kidney cells during conditions with Ischemic-Reperfusion (I/R) induced acute kidney injury (AKI), bearing in mind that mPTP opening is decisive for acute kidney damage. Glycogen synthase kinase (GSK)-3β is involved in signaling pathways for mitochondrial permeability transition (mPT), which is why the derivative Thiadiazolidinone-8 (TDZD-8) has been used for the recovery of kidney cells during AKI [115]. Another possibility for the treatment of AKI is using the terapeptide SS-31 Szeto-Sciller peptide or Bendavia because it stabilizes cardiolipin by controlling the activity of Cytochrome C (cyt C) and inhibiting the mitochondrial permeability transition (mPT), which improves kidney perfusion. The antioxidant role of elamipretide (SS-31) has been documented in the mitochondria of heart and kidney cells by restoring redox homeostasis by improving mitochondrial aerobic respiration. A representative of the ROS metabolism and ATP synthesis modulator uses a new pathophysiological mechanism for kidney and heart protection. Mitochondrial acid 5 (MA-5) is a derivative of indole acetic acid. Its role is to increase the production of ATP, prevent fragmentation, and preserve the dynamics of mitochondria. Activator of Mitochondria Biogenesis is also important for the reduction of oxidative stress and the recovery of kidney cells in acute damage, and it is possible to achieve this with the help of a group of agents still in the phase of experimental testing [117,118].

This effect is achieved with the help of Resveratrol, which is otherwise created by some plants to protect against fungal infection. In the case of acute lung damage caused by sepsis, it increases the respiratory capacity of mitochondria by increasing the mass of mitochondria and suppressing inflammation caused by macrophages. Adenosine monophosphate protein kinase (AMPK) and activator 5-amino imidazole-4-carboxamide ribonucleotide (AICAR) were examined for the treatment of acute renal proximal tubule damage caused by cisplatin. Induction of mitochondrial biogenesis was also achieved with formoterol, an agonist of β2-adrenoreceptor because it helps to increase the number of copies of mt DNA and thus oxygen consumption in the cells of the proximal kidney tubules [119,120].

To reduce oxidative stress in mitochondria, mitochondria-targeted antioxidants (MitoQ) are used antioxidants, i.e., ROS capture compounds, which use lipophilic cations for transport, pass the lipid membrane in a membrane potential-dependent manner. TMA conjugates with triphenyl alkyl phosphonium cation (TPP+) in the mitochondrial matrix. Their main disadvantage is that TPP-linked antioxidants damage the inner mitochondrial membrane during their accumulation and reduce ATP synthesis and aerobic metabolism [121,122].

The appropriate application of antioxidants is of paramount importance in mitochondrial research.

Redox signaling pathways in heart and kidney cells are crucial, and excessive or inappropriate use of antioxidants can lead to compensatory up-regulation of mitogen-activated protein kinase (MAPK) pathways, potentially causing a breakdown of the endogenous antioxidant system. This underscores the need for responsible and cautious research practices, including practical pharmacokinetics and the specific accumulation of antioxidants in cells, which is the essence of trials related to the recovery of mitochondrial function [123].

Exciting progress is being made in experimental and preclinical research related to levosimendan. This compound, a mitochondrial K (ATP) channel opener, shows promise in improving kidney perfusion. It achieves this through selective vasodilatory mechanisms via ATP-sensitive K+ channels and inhibition of Phosphodiesterase-3 enzyme. These findings inspire hope for the potential future applications of levosimendan in improving kidney function [124,125,126,127,128].

## 3. Conclusions

The pathophysiological mechanism that is induced by oxidative stress and is consequently responsible for the onset of acute cardiorenal syndrome is analyzed in this paper. The severity of the induced kidney damage cannot be predicted, nor can the activation of interdependent processes by which the impaired kidney function affects the heart function. This is important because there is evidence that the risk factors present in the different stages of acute kidney injury are the same as those for chronic kidney injury and heart injury. After all, severe kidney injury is a major accelerator of the induction and future progression of heart injury.

Initiating regenerative processes to recover tubule epithelial cells after acute kidney damage is also important. That is why some knowledge about the impact of mitochondrial function and stress on the function of heart and kidney cells was analyzed. The concept of mitohormesis indicates that the level of stress and the duration of the stress are important for triggering the hormetic response. Disturbance of function, perturbations of mitochondria, and resistance of mitochondria to stress have an impact on kidney and heart function.

The aim of prevention and adequate treatment in cases of acute kidney damage is to achieve a reversible outcome and prevent complications with potential damage to other organs. Because of that, the paper highlighted up to now knowledge related to agents that affect mitochondrial stress and the balance of mitochondrial proteins and oxidative stress. They have the potential to postpone kidney and heart damage through their action on oxidative stress as an etiopathogenetic mechanism for the emergence of acute cardiorenal syndrome, and in this way, they lead us to a new therapeutic solution for this syndrome.

The exact mechanism of cardiorenal syndrome type 3 is complex and not yet understood. There is a link between acute kidney injury and the pathophysiology of developing cardiovascular disease. We showed a greater understanding of the development of cardiorenal mechanisms. Acute kidney injury is associated with acute and chronic cardiovascular complications. This kind of interaction is possible in different ways and is very dynamic. Further research in the area of risk mechanisms and control of oxidative stress will improve the management of acute cardiorenal syndrome.

## Figures and Tables

**Figure 1 diagnostics-14-02094-f001:**
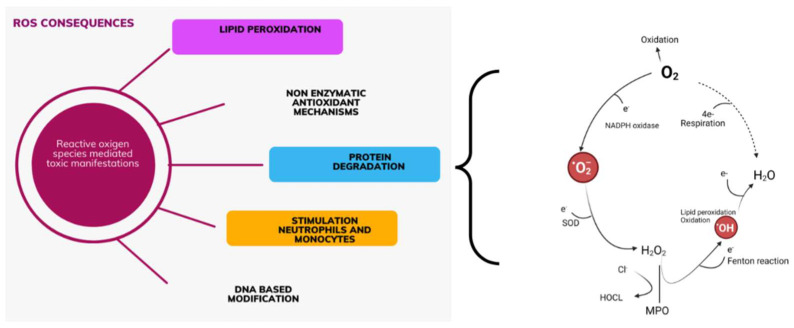
ROS-induced pathways in the kidney and heart relationship (created with BioRender.com on 7 July 2024). ROS are a normal outcome of mitochondrial respiration and energy production. They are highly reactive molecules with toxic potential. Production ROS affects many signaling pathways simultaneously. Nonradicals: H_2_O_2_—hydrogen peroxide; NADPH—Nicotinamide adenine dinucleotide phosphate; SOD-superoxide dismutase; H_2_O—water; MPO—myeloperoxidase; Cl^−^—Chlorine; HOCL-Hypochlorous acid; O_2_—oxygen reduction reaction; e^−^—electron; Radicals: O_2_^−^—superoxide anion; OH^−^—Hydroxide anion.

**Figure 2 diagnostics-14-02094-f002:**
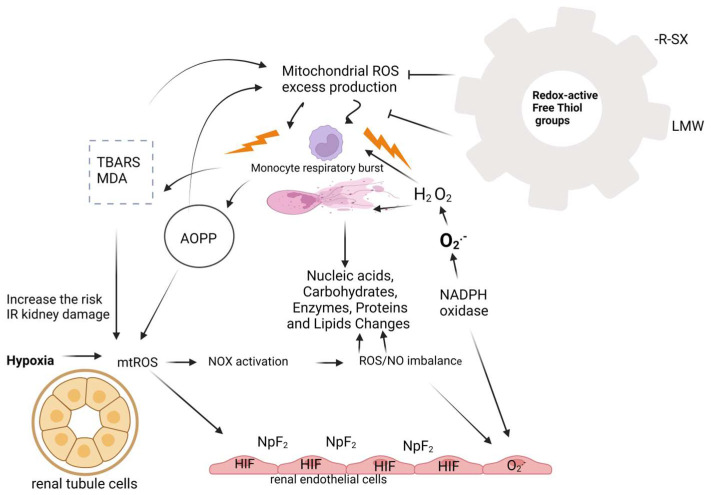
Oxidative stress and AKI (created with BioRender.com on 7 July 2024). The effect of oxidative stress on kidney tubule cells is shown in the figure. Changes in the structure and function of kidney mitochondria during AKI cause a disturbance in the energy metabolism in the kidney cells. The mutual interaction between different structures of the kidney and oxidative stress is based on the number of mitochondria, the way of production of reactive oxygen species, and the regulation of the response to oxidative stress. Downward black arrows: induction; black T: inhibition; yellow thunderbolt: oxidative damage. ROS—reactive oxygen species; TBARS—thiobarbituric acid; MDA—malonyl dialdechyde; AOPP—products of oxidative modification protein; R-SX—thiol sulfhydryl group; LMW—low molecular weight; NO—nitrogen oxide; NADPH—Nicotinamide adenine dinucleotide phosphate; HIF—hypoxia-inducible factors; NpF_2_—nuclear factor erythroid 2-related factor 2; NOX—Nicotinamide adenine dinucleotide phosphate oxidase; H_2_O_2_—hydrogen peroxide; O_2_^−^—superoxide anion.

**Figure 3 diagnostics-14-02094-f003:**
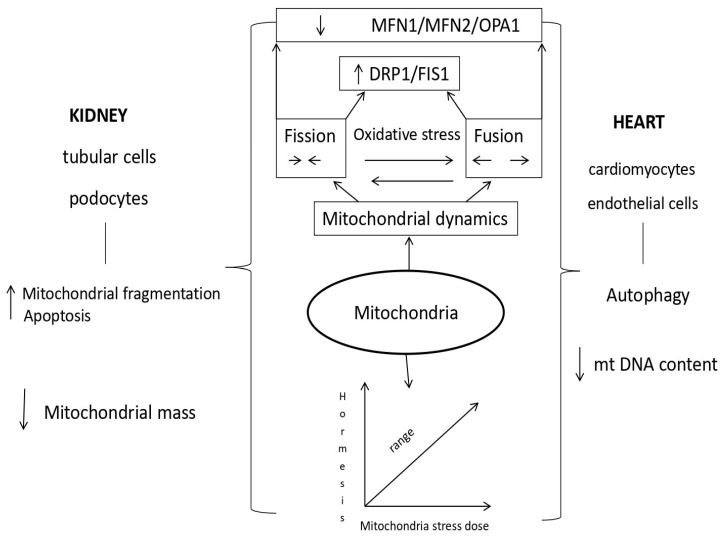
Mithochondrial ROS in cardiorenal syndrome type 3. Mechanisms of reactive oxygen species generation in mitochondria, kidney, and heart cells during AKI are shown in the figure. Mitochondria are the main source of reactive oxygen species and are directly responsible for the intensification of oxidative stress in AKI as well as for triggering mechanisms that are responsible for remodeling and progression of heart damage. Upward black arrows: increase; downward black arrows: decrease; black Rightwards Arrow Over Leftwards Arrow: equilibrium arrow. MFN1: Mitofusin 1; MFN2: Mitofusin 2; OPA1: Optic atrophy 1; DRP: Dyamin-related protein 1; FIS1: Mitochondrial fission 1 protein; mtDNA: Mitochondrial double-stranded deoxyribonucleic acid.

**Table 1 diagnostics-14-02094-t001:** Preclinical studies targeting DRP1 link Kidney and/or Heart.

	Experimental Setting	Target	Main Finding	Reference
Kidney	Ltbp4S-/- mouse model	AKI-IR/LTBP4	AKI to CKD transition via DRP1 pathways	[70]
Kidney/Heart	Renal ischemia/ Doxorubicin rats	Systemic inflammation/heart damage after renal ischemia	↑ DRP1	[71]
Kidney	Aldosterone-induced podocytes injury/mouse/cultured podocytes	P53/DRP1 mitochondrial dysfunction	↑ DRP1	[72]
Heart	Neonatal murine cardiomyocytes/adult rat hearts	Activation of DRP1 by myocardial IR and LV impairment	DRP1 mediated diastolic dysfunction and therapeutic benefits DRP1 inhibition	[73]
Heart	In vivo mice models of septic cardiomyopathy/cardiac cell line model	Inhibition DRP1/FIS1	Treatment P110 reduced cardiac mitochondrial fragmentation and improved ATP production	[74]
Kidney	In vivo genetic murine models in wild-type mice	IR/therapeutic potential DRP1 deletion in proximal tubule epithelium before IR	Protects against kidney injury	[75]
Acute cardiorenal syndrome	In vivo mice IR models/in vitro cell	IR by bilateral renal artery clamping	Cardiac dysfunction induced by renal IR improved by DRP1 inhibition	[76]
Heart	Tamoxifen-inducible/DRP1-hetCKO mice	Role endogenous DRP1 in the protection LV function against IR	Left ventricular dysfunction and increased susceptibility to IR	[77]

Abbreviations: DRP1—Dynamin-related protein 1; ↑—high level; FIS1—mitochondrial fission 1 protein; IR—ischemia-reperfusion; ATP—Adenosine triphosphate; LV—Left Ventricle Heart; AKI—acute kidney injury; LTBP4—Latent Transforming Growth. Factor Beta Binding protein 4; CKD—Chronic kidney disease; DRP1-hetCKO—cardiac-specific DRP1 heterozygote knockout mice.

## Data Availability

No new data were created or analyzed in this study. Data sharing is not applicable to this article.

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
