# Peer review of "The Role of Oxidative Stress as a Mechanism in the Pathogenesis of Acute Heart Failure in Acute Kidney Injury"

_diagnostics, 2024, doi:10.3390/diagnostics14182094_

Round 1
Reviewer 1 Report (Previous Reviewer 4)
Comments and Suggestions for Authors
Albeit improved and implementing a more focused paragraph on the review topic, the current version of the manuscript is still quite dispersive; a lot of words are spent on molecular mechanisms of mitochondrial damage; however, across the review, it is unclear the role of all these molecular pathways until the "Discussion" (quite unusual for a review paper). In particular, mechanism are described, but the role of each one and the relationship with clinical conditions is very difficult to understand for the reader. Moreover, the quality of written English make it very difficult to follow and understand.
Comments on the Quality of English LanguageQuality of written English is still not acceptable; there are many convoluted sentences, redundancies, and improper expressions. A massive native speaker revision is necessary.
Author Response
For review article
|
Re-Response to Reviewers Comments
|
|
||||
|
1. Summary
|
|
|
|
||
|
Dear Editor and Diagnostics Editoral Office, Thank you very much for taking the time to review this manuscript: “THE ROLE OF OXIDATIVE STRESS AS A MECHANISM IN THE PATHOGENESIS OF HEART FAILURE IN ACUTE KIDNEY INJURY”; Manuscript ID: diagnostics-3034354, by Danijela Tasic and Zorica Dimitrijevic. Please find the detailed re-respons below and the corresponding corrections in red letters therefore you can track changes in the re-submitted files. In addition, the list of changes and answers to the reviewers questions are given below.
|
|
||||
|
|
|
||||
|
2. Point-by-point response to Comments and Suggestions for Authors |
|
|
|||
|
Reviewer 1:Comments 1: Albeit improved and implementing a more focused paragraph on the review topic, the current version of the manuscript is still quite dispersive; a lot of words are spent on molecular mechanisms of mitochondrial damage; however, across the review, it is unclear the role of all these molecular pathways until the "Discussion" (quite unusual for a review paper). In particular, mechanism are described, but the role of each one and the relationship with clinical conditions is very difficult to understand for the reader. Moreover, the quality of written English make it very difficult to follow and understand.
|
|
|
|||
|
Response 1: We would like to thank the reviewer for his effort in analyzing our paper. We tried to focus on the molecular mechanisms of oxidative stress that are induced in kidney cells during acute kidney damage and that further induce first functional and then structural deterioration of kidney cells and then the heart in acute cardiorenal syndrome type 3. The production of oxidants and antioxidants can be disrupted by a chain of new reactions triggered by oxidative stress, despite the strict regulation of the process. Initial damage in the early stages occurs at the subcellular level. Mitochondria, as a significant source of antioxidants and as a site of enzyme activity, were analyzed separately. The aim of the molecular mechanisms described in detail is to explain the occurrence of disorders of compensatory mechanisms. These are compensatory mechanisms whose purpose is to maintain the perfusion of the kidneys and heart as vital organs. The final effect of oxidative stress as a mechanism in the later stages of kidney and heart failure is not the subject of this paper. The discussion in this paper means that researching the molecular mechanisms of oxidative stress has its application, which is debatable and subject to checks and changes, according to current knowledge.
|
|
|
|||
|
Comments 2: Quality of written English is still not acceptable; there are many convoluted sentences, redundancies, and improper expressions. A massive native speaker revision is necessary.
|
|
|
|||
|
Response 2: Thank you for pointing this out. We are grateful for the suggestion. The editor who is a native English speaker has corrected the current version of the paper several times, but from the American-speaking area. This time, we included a proofreader from the UK who reviewed the paper.
|
|
|
|||
Reviewer 2:Comments 1: Scheme 1. I could not see ROS induced pathways in the kidney and heart relationship. Where are the kidney and heart?
Response 1: Thank you for pointing this out. Scheme 1 shows the mechanisms of oxidative stress and the chain of reactions that have their roles in the stages of the adaptation process in both organs. The goal of this scheme is to show that the production of ROS in the cells of these organs is a very sophisticated and strictly regulated process. That is why the presentation is given in this way.
Comments 2: I don't understand the term "renal endothelial cells". Do you mean "Glomerular Endothelial Cells" or " tubular epithelial cells" ? I could not see the so called AKI model here. IR? Ischemia/reperfusion injury-related acute kidney injury (IRI-AKI) ?
Response 2: Thank you for pointing this out. Epithelial cells of the tubules are in close contact with the microvasculature of the kidney, so the influence of endothelial cells on the function of the epithelial cells of the kidney has been studied a lot, as well as the connection and interdependence of these two types of cells. Disruption of the endothelial function at the microcirculation level and the activation of the epithelial cells of the tubules, which were created under the influence of various signaling pathways, are of great importance for the creation of a vicious circle.
Preclinical studies on the AKI model are shown in Table 1.
Comments 3: Mithochondrial ROS in cardiorenal syndrome (CRS) "type 3."
If you focused on cardiorenal syndrome type 3, the title should be revise as follows: The Role of Oxidative Stress as a Mechanism in the 2 Pathogenesis of "Acute" Heart Failure in Acute Kidney Injury. My problem is, how do authors define four to five types of CRS for the mechanism/ mediators of oxidative injury? Did mithochondrial responses differ in various types of cardiorenal interaction? I think they are quite similar.
The English is very difficult to understand/incomprehensible.
Response 3: We thank the reviewer for the time invested in the analysis of the paper and suggestions. The title has been corrected with the word acute, which is colored blue.
In relation to the type of adaptive response and cell homeostasis, the biochemical characteristics of oxidative stress parameters differ, making their determination a useful marker for monitoring the occurrence, degree of damage and progression of kidney disease. Increase in oxidative stress parameters is not only a sign of the activity of the oxidative process but also a consequence of their accumulation. The heart, as an organ with higher metabolic requirements compared to the kidney, significantly increases the consumption of enzymatic and non-enzymatic antioxidant potential. The remodeling process is initially adaptive in nature with the aim of preserving muscle mass and normalizing the heart wall, which enables further adaptation of the heart to the load and demands.
Thank you for pointing this out. We are grateful for the suggestion. The editor who is a native English speaker has corrected the current version of the paper several times, but from the American-speaking area. This time, we included a proofreader from the UK who reviewed the paper.
Finally,
We would like to thank once again the referees for their valuable comments and suggestions, which improved our manuscript.
Thank you very much.
With respect and gratitude,
Danijela Tasic and Zorica Dimitrijevic
University of Nis
Faculty of Medicine
UCC Nis, Clinic of nephrology “Prof. dr Spira Strahinjic”
Reviewer 2 Report (Previous Reviewer 2)
Comments and Suggestions for Authors
1.Scheme 1. I could not see ROS induced pathways in the kidney and heart relationship. Where are the kidney and heart?
2.I don't understand the term "renal endothelial cells". Do you mean "Glomerular Endothelial Cells" or " tubular epithelial cells" ? I could not see the so called AKI model here. IR? Ischemia/reperfusion injury-related acute kidney injury (IRI-AKI) ?
3.Scheme 3. Mithochondrial ROS in cardiorenal syndrome (CRS) "type 3."
If you focused on cardiorenal syndrome type 3, the title should be revise as follows: The Role of Oxidative Stress as a Mechanism in the 2 Pathogenesis of "Acute" Heart Failure in Acute Kidney Injury. My problem is, how do authors define four to five types of CRS for the mechanism/ mediators of oxidative injury? Did mithochondrial responses differ in various types of cardiorenal interaction? I think they are quite similar.
Comments on the Quality of English LanguageThe English is very difficult to understand/incomprehensible.
Author Response
For review article
|
Re-Response to Reviewers Comments
|
|
||||
|
1. Summary
|
|
|
|
||
|
Dear Editor and Diagnostics Editoral Office, Thank you very much for taking the time to review this manuscript: “THE ROLE OF OXIDATIVE STRESS AS A MECHANISM IN THE PATHOGENESIS OF HEART FAILURE IN ACUTE KIDNEY INJURY”; Manuscript ID: diagnostics-3034354, by Danijela Tasic and Zorica Dimitrijevic. Please find the detailed re-respons below and the corresponding corrections in red letters therefore you can track changes in the re-submitted files. In addition, the list of changes and answers to the reviewers questions are given below.
|
|
||||
|
|
|
||||
|
2. Point-by-point response to Comments and Suggestions for Authors |
|
|
|||
|
Reviewer 1:Comments 1: Albeit improved and implementing a more focused paragraph on the review topic, the current version of the manuscript is still quite dispersive; a lot of words are spent on molecular mechanisms of mitochondrial damage; however, across the review, it is unclear the role of all these molecular pathways until the "Discussion" (quite unusual for a review paper). In particular, mechanism are described, but the role of each one and the relationship with clinical conditions is very difficult to understand for the reader. Moreover, the quality of written English make it very difficult to follow and understand.
|
|
|
|||
|
Response 1: We would like to thank the reviewer for his effort in analyzing our paper. We tried to focus on the molecular mechanisms of oxidative stress that are induced in kidney cells during acute kidney damage and that further induce first functional and then structural deterioration of kidney cells and then the heart in acute cardiorenal syndrome type 3. The production of oxidants and antioxidants can be disrupted by a chain of new reactions triggered by oxidative stress, despite the strict regulation of the process. Initial damage in the early stages occurs at the subcellular level. Mitochondria, as a significant source of antioxidants and as a site of enzyme activity, were analyzed separately. The aim of the molecular mechanisms described in detail is to explain the occurrence of disorders of compensatory mechanisms. These are compensatory mechanisms whose purpose is to maintain the perfusion of the kidneys and heart as vital organs. The final effect of oxidative stress as a mechanism in the later stages of kidney and heart failure is not the subject of this paper. The discussion in this paper means that researching the molecular mechanisms of oxidative stress has its application, which is debatable and subject to checks and changes, according to current knowledge.
|
|
|
|||
|
Comments 2: Quality of written English is still not acceptable; there are many convoluted sentences, redundancies, and improper expressions. A massive native speaker revision is necessary.
|
|
|
|||
|
Response 2: Thank you for pointing this out. We are grateful for the suggestion. The editor who is a native English speaker has corrected the current version of the paper several times, but from the American-speaking area. This time, we included a proofreader from the UK who reviewed the paper.
|
|
|
|||
Reviewer 2:Comments 1: Scheme 1. I could not see ROS induced pathways in the kidney and heart relationship. Where are the kidney and heart?
Response 1: Thank you for pointing this out. Scheme 1 shows the mechanisms of oxidative stress and the chain of reactions that have their roles in the stages of the adaptation process in both organs. The goal of this scheme is to show that the production of ROS in the cells of these organs is a very sophisticated and strictly regulated process. That is why the presentation is given in this way.
Comments 2: I don't understand the term "renal endothelial cells". Do you mean "Glomerular Endothelial Cells" or " tubular epithelial cells" ? I could not see the so called AKI model here. IR? Ischemia/reperfusion injury-related acute kidney injury (IRI-AKI) ?
Response 2: Thank you for pointing this out. Epithelial cells of the tubules are in close contact with the microvasculature of the kidney, so the influence of endothelial cells on the function of the epithelial cells of the kidney has been studied a lot, as well as the connection and interdependence of these two types of cells. Disruption of the endothelial function at the microcirculation level and the activation of the epithelial cells of the tubules, which were created under the influence of various signaling pathways, are of great importance for the creation of a vicious circle.
Preclinical studies on the AKI model are shown in Table 1.
Comments 3: Mithochondrial ROS in cardiorenal syndrome (CRS) "type 3."
If you focused on cardiorenal syndrome type 3, the title should be revise as follows: The Role of Oxidative Stress as a Mechanism in the 2 Pathogenesis of "Acute" Heart Failure in Acute Kidney Injury. My problem is, how do authors define four to five types of CRS for the mechanism/ mediators of oxidative injury? Did mithochondrial responses differ in various types of cardiorenal interaction? I think they are quite similar.
The English is very difficult to understand/incomprehensible.
Response 3: We thank the reviewer for the time invested in the analysis of the paper and suggestions. The title has been corrected with the word acute, which is colored blue.
In relation to the type of adaptive response and cell homeostasis, the biochemical characteristics of oxidative stress parameters differ, making their determination a useful marker for monitoring the occurrence, degree of damage and progression of kidney disease. Increase in oxidative stress parameters is not only a sign of the activity of the oxidative process but also a consequence of their accumulation. The heart, as an organ with higher metabolic requirements compared to the kidney, significantly increases the consumption of enzymatic and non-enzymatic antioxidant potential. The remodeling process is initially adaptive in nature with the aim of preserving muscle mass and normalizing the heart wall, which enables further adaptation of the heart to the load and demands.
Thank you for pointing this out. We are grateful for the suggestion. The editor who is a native English speaker has corrected the current version of the paper several times, but from the American-speaking area. This time, we included a proofreader from the UK who reviewed the paper.
Finally,
We would like to thank once again the referees for their valuable comments and suggestions, which improved our manuscript.
Thank you very much.
With respect and gratitude,
Danijela Tasic and Zorica Dimitrijevic
University of Nis
Faculty of Medicine
UCC Nis, Clinic of nephrology “Prof. dr Spira Strahinjic”
This manuscript is a resubmission of an earlier submission. The following is a list of the peer review reports and author responses from that submission.
Round 1
Reviewer 1 Report
Comments and Suggestions for Authors
The review is well written and the author has summarized and discussed the role of oxidative stress in the heart failure resulting from acute kidney injury.
Minor comments:
Please do spelling checks especially for
Relationchip to relationship in Scheme 1
Cradiorenal to cardiorenal in Scheme 2
Comments on the Quality of English LanguageFew minor spelling mistakes
Reviewer 2 Report
Comments and Suggestions for Authors
Dear editors:
It is a great honor and pleasure for me to be invited as the reviewer for this important work. Tasić Danijela reviewed the role of ROS in the Pathogenesis of Heart Failure in Acute Kidney Injury. I have a number of comments concerning this study:
1. The co-existence of Mechanism and Pathogenesis in the title is redundant. In light of the scope of the article (e.g., Scheme 2. Mithochondrial Dysfunction), the term of mitochondrial ROS could be considered.
2. Line 20: Compared with “renocardial syndrome”, “renocardiac syndrome” is the more common form. However, the term of cardiorenal syndrome is the most common form in the field of cardiorenal interaction. Please uniform the terms in the article. Indeed, the only term “cardiorenal syndrome” was used in the reference list. Likewise, KDIGO is the more common form than K/DIGO.
3. Line 25; Since the abbreviation of AKI was used, “acute kidney injury” should be replaced with AKI in Line 29 and other parts in the text. The authors should provide an abbreviation list at the end of the article.
4. The introduction seems shallow and redundant. The traditional cardiorenal interaction has been well-known. The novel hypothesis of ROS/ RNS overproduction derived from mitochondrial dysfunction in the crosstalk between cardiomyocytes and renal tubule cells should be strengthened.
5. Line 90: Their measurement is also a marker of monocyte activation (scheme 1)[41,42]. However, I could not see the monocyte activation mentioned in the section 1.1 in the scheme 1. I think the quality of scheme 1 is suboptimal that should be remade in light of the section 1.1.
6. There are too many grammar errors and misspelled English words, e.g., Scheme 1. Role ROS in the kidney and heart relationchip.; Scheme 2. Mithochondrial Dysfunction in “Cradiorenal” Syndrome type 3. => Cardiorenal. Too many errors limit the scientific merit of the manuscript.
7. Figure legends should be provided in the schemes. I could not understand the meaning of the schemes.
8. After carefully reading the article, I think there is still a huge gap between renal oxidative stress and heart failure. How could renal mtROS/ mtRNS in AKI affect myocardiocytes that result in heart failure? Given cardiorenal syndrome type 3and type 4 differ, what is the difference of mtROS/ mtRNS production between AKI and CKD?
Comments on the Quality of English LanguageExtensive editing of English language is required.
Reviewer 3 Report
Comments and Suggestions for Authors
The manuscript title THE ROLE OF OXIDATIVE STRESS AS A MECHANISM IN THE PATHOGENESIS OF HEART FAILURE IN ACUTE KIDNEY INJURY showed to oxidative stress as a potential contributor for heart damage during AKI. This manuscript is interesting because intent to explain the participation of mitochondria in acute. heart faliure after AKI. However, a lack of information describing the possible mechanisms that enhance heart failure or that link the processes that lead to it make this review very ambiguous. Perhaps the author could include an informative table suggesting possible links between AKI and heart failure. The data presented is rather a descriptive description of information that does not provide anything new.
minor comments
Figure 1 is very small and the letters cannot be seen
there are grammatical errors
Reviewer 4 Report
Comments and Suggestions for Authors
In this paper, the Author reviews mechanisms of oxadative stress and their role in type 3 cardiorenal syndrome.
MAIN COMMENT
In general, one of the main issues with the paper is the quality of written English: many sentences show a convoluted construction, with unusual expressions and repeated concepts, making the review very difficult to read. It is, thus, warranted a general and a native speaker revision before considering it for publication.
Another main problem are the references: out of the first 44, at least 14 are narrative reviews (guidelines and systematic reviews not included). This is not acceptable, since various statements should cite the original studies with experimental evidence. It is ok to cite few narrative reviews, however their number should be kept as low as possible.
Last but not least, the part of the paper actually focused on the subject of the review is relatively small (just the last paragraph, since all the previous are about oxidative stress in general, with few recalls to heart and kidney failure). I would advise to edit the manuscript, cutting unnecessary details and repeated concepts in the previous section and developing the last one, possibly expanding the amount of evidence-supported statement (i.e., results from an experimental study such as https://doi.org/10.1155/2020/1605358 have not been included; other experimental studies have explored the topic and should be reviewed). The conclusions are quite puzzling as the role of oxidative stress (main topic of the review) is not mentioned per se if not in the context of it being under-researched. The rest of the section is made up of very generic and archetypical sentences. Please revise.
OTHER COMMENTS
- Introduction, page 1, lines 24 to 27: this is a quite unusual definition for AKI, also not taking into account the case of acute over chronic. Please revise.
- Introduction, first paragraph: in general, this paragraph is quite redundant.
- Lines 40-45: Definitions of heart failure and acute decompensated heart failure causality according to 2021 ESC are inaccurate and misleading. Please revise.
- Lines 50-53: By using “acute kidney injury”, the author implies the KDIGO definition which is independent from etiology and identifies serum creatinine as the only biomarker for staging. Please revise.
- Please define acronyms at their first use (i.e. TBARS, AOPP, etc.)
- Please check reference 52: in the cited paper HIF or NpF2 are not mentioned.
- “The main factor for maintaining physiological tone in the blood vessels of the kidneys, endogenous O2, is induced by NADPH oxidase (NOX) in the blood vessels”: please provide reference.
- “The concentration of MDA in the urine is directly proportional to the degree of kidney tubule damage, which is why it can be a good indicator of tubule damage in acute renal failure.”: please provide reference.
- Despite a role for DRP1 both in kidney and heart failure, is there experimental evidence for a role of DRP1 in heart during AKI (which should be the topic)?
- Scheme 1 caption: Please correct misspellings.
- Scheme 2: please correct spelling mistakes (mitohondria, fussion, etc.)
- Line 169 to end of page. Please define AOB, add citation and/or divide paragraphs more clearly to understand references.
Comments on the Quality of English LanguageSee comments to Authors
Round 2
Reviewer 2 Report
Comments and Suggestions for Authors
Figure legends should be provided in the scheme 2 and 3, not merely the abbreviations.
The resolution of the scheme 1 should be refined.
Abbreviations should be organized from a to z.
superoxide anion: O2•−
